# Modeling of MEMS Transducers with Perforated Moving Electrodes

**DOI:** 10.3390/mi14050921

**Published:** 2023-04-24

**Authors:** Karina Šimonová, Petr Honzík

**Affiliations:** Faculty of Transportation Sciences, Czech Technical University in Prague, Konviktská 20, 110 00 Praha, Czech Republic; abramkar@fd.cvut.cz

**Keywords:** analytical modeling, electroacoustic transducers, MEMS microphones, perforated plate

## Abstract

Microfabricated electroacoustic transducers with perforated moving plates used as microphones or acoustic sources have appeared in the literature in recent years. However, optimization of the parameters of such transducers for use in the audio frequency range requires high-precision theoretical modeling. The main objective of the paper is to provide such an analytical model of a miniature transducer with a moving electrode in the form of a perforated plate (rigid elastically supported or elastic clamped at all boundaries) loaded by an air gap surrounded by a small cavity. The formulation for the acoustic pressure field inside the air gap enables expression of the coupling of this field to the displacement field of the moving plate and to the incident acoustic pressure through the holes in the plate. The damping effects of the thermal and viscous boundary layers originating inside the air gap, the cavity, and the holes in the moving plate are also taken into account. The analytical results, namely, the acoustic pressure sensitivity of the transducer used as a microphone, are presented and compared to the numerical (FEM) results.

## 1. Introduction

Currently, the vast majority of MEMS microphones production, increasing rapidly in recent years, uses the electrostatic principle of electroacoustic transduction [1] (although piezoelectric types exist [2]). Such devices consist of moving electrodes of circular [3], square [4,5], or other [6] shapes and perforated single [3] or double backplates [7,8]. Note that such MEMS structures can be employed in other domains than audio, such as energy transfer, energy harvesters, and resonators [9,10,11]. However, new designs presenting technological advances have been proposed recently in the literature, such as a microphone with moving microbeam [12], or transducers (sources and microphones) with perforated moving electrodes. The mean motivation for the work presented herein is the latter case with electrodes in the form of elastic perforated plates clamped at all boundaries [13] or rigid elastically supported perforated plates [14,15,16,17]. Although these experimental studies contain approximate theoretical models, mainly based on the lumped elements approach, the precise analytical modeling is still of high interest.

In order to provide high-precision results on sensitivity and bandwidth, the models of electroacoustic transducers (miniaturized or not) should take into account the damping effects of the viscous and thermal boundary layers originating in the narrow regions such as the air gap between the moving and fixed electrodes. The strong coupling between the displacement field of the moving electrode and the acoustic field inside the transducer should be also accounted for when appropriate. In addition to these effects, the model of the transducer with a perforated moving electrode has to deal with the acoustic short circuit between the incident acoustic pressure and the pressure field inside the transducer caused by the perforation. This leads to the sensitivity roll-off at lower frequencies, which has to be calculated correctly when precise theoretical modeling allowing the optimization of the transducer behavior in the audio frequency range is required.

While the precise models of the transducers with perforated moving electrodes are still missing, to our knowledge, several models taking into account the perforation of the fixed electrodes and the acoustic short circuit can be found in the literature. The classical lumped-element models of condenser microphones, such as [18], use the “porosity” approach; the more recent lumped-element model [19] deals with acoustic short circuit through the venting hole. With regard to more advanced models, ref. [20] and, more recently, [21,22] took into account the effects of holes in the fixed electrode accounting for the position of the holes, and ref. [23] employed the impedance approach. Vibration of a very thin perforated backplate of an MEMS transducer was taken into account in [24]. In [25], the effect of the acoustic short circuit through thin slits surrounding the moving electrode in the form of a microbeam was included in the complex wavenumber for the acoustic pressure in the air gap. In the same reference, the acoustic pressure in the air gap was expressed using integral formulation with appropriate Green’s function, which was not expressed as a series expansion over the eigenfunctions of the moving electrode. Such a fomulation is also advantageous in the case of rectangular geometries [26,27] and is therefore used herein. It is worth mentioning the numerical methods, namely, the finite element method, which can take into account the thermoviscous losses and the coupling effects without geometry-dependent approximations [28]. However, numerical methods generally suffer from high computational costs, compared to analytical methods, and are usually used as a reference against which the analytical results can be tested.

The present paper deals mainly with the theoretical modeling of the acoustic field inside a miniaturized electroacoustic transducer with a square perforated moving electrode, taking into account its coupling with the vibration of the moving electrode, the acoustic short circuit through the perforation, and the thermoviscous losses originating in the narrow regions inside the transducer. Two types of the perforated moving electrode are considered: (i) the rigid elastically supported square plate, partially inspired by [14] (see Figure 1a), and (ii) the flexible square plate clamped at all boundaries, partially inspired by [13] (see Figure 1b).

Section 2 presents viscous effects in short narrow holes and governing equations for the acoustic pressure field in the thin air gap between the perforated moving electrode and the fixed one (backplate) using the porosity approach. Then, the solutions for the acoustic pressure are expressed for the case of uniform (piston-like) and nonuniform movement of the moving electrode, corresponding to the rigid elastically supported and flexible plate, respectively. The coupling of the acoustic field with vibration of the plates of both types, leading to the expression of their displacements, is finally derived, with the eigenfunctions of the perforated flexible clamped plate being given approximately in Appendix A. In Section 3, the analytically calculated acoustic pressure sensitivities of the transducers used as microphones are depicted and compared with the numerical (FEM) results. The influence of some geometrical parameters is discussed. This section is followed by the conclusion in Section 4.

## 2. Analytical Solution

In this section, the analytical solution of the problem is expressed in frequency domain (the time dependence being $e^{j\omega t}$; ω is the angular frequency). The acoustic field inside the transducer and the displacement of the moving electrode is searched for as a response to harmonic incident acoustic pressure $p_{inc}$ (assumed to be uniform over the moving electrode surface).

### 2.1. Description of the Device

The device consists of a moving electrode in the form of a square perforated plate of side 2a and thickness $h_{p}$ with *N* square holes of side $a_{h}$, with the air gap between the plate and the backplate of thickness $h_{g}$ surrounded by a peripheral cavity described by its volume $V_{c}$ and acoustic impedance $Z_{c}$ (see Figure 2). The perforation ratio $R=Na_{h}^{2}/\left(4a^{2}\right)$ is the ratio of total surface occupied by the holes and the area of the plate. In the case of a moving electrode in the form of a rigid elastically supported square plate (Figure 1a), the plate displacement is uniform and the cavity is connected with the incident acoustic pressure $p_{inc}$ through slits of thickness $h_{s}$ along the arms supporting the plate.

### 2.2. The Acoustic Pressure Field inside the Transducer

The system is supposed to be filled with thermoviscous fluid (air in this case) with the following properties: the density $\rho_{0}$, the adiabatic speed of sound $c_{0}$, the heat capacity at constant pressure per unit mass $C_{p}$, the specific heat ratio γ, the shear viscosity coefficient μ, and the thermal conduction coefficient $\lambda_{h}$. Since the air gap thickness $h_{g}$ is supposed to be much smaller than other dimensions of the air gap and smaller than the wavelengths considered, even at high frequencies, the acoustic pressure in the air gap is assumed to depend on the x,y spatial coordinates only and denoted $p_{g}\left(x,y\right)$. The particle velocity and temperature variation (that generally depend on the *z* axis due to the viscous and thermal boundary layers effects) are then replaced by their mean values over the air gap thickness. The acoustic pressure in the cavity volume $p_{c}$ is supposed to be uniform. The displacement of the moving electrode is denoted ξ.

#### 2.2.1. Viscous Effects Originating in the Holes in the Perforated Moving Electrode

For the sake of simplicity, the holes in the moving electrode are supposed to have circular cross-section instead of the square one, with the radius of the equivalent cylindrical hole being given by $R_{h}=a_{h}/\sqrt{\pi }\;$ (thus $R=\frac{N\pi R_{h}^{2}}{4a^{2}}$), see Figure 3. The particle velocity $v_{z}\left(r,z\right)$ in such a hole is governed by the diffusion equation [29]
(1)$$\left(\frac{1}{r}\frac{\partial }{\partial r}r\frac{\partial }{\partial r}+k_{v}^{2}\right)v_{z}\left(r,z\right)=\frac{1}{\mu }\frac{\partial }{\partial z}p\left(z\right),$$
with the diffusion wavenumber
(2)$$k_{v}=\frac{1-j}{\sqrt{2}}\sqrt{\frac{\omega \rho_{0}}{\mu }},$$
with *j* being the imaginary unit, and subjected to the nonslip boundary condition at $r=R_{h}$
(3)$$v_{z}\left(R_{h},z\right)=j\omega \xi \left(x,y\right).$$

The velocity of the moving electrode jωξ(x,y) at the position of the hole is assumed to be approximately uniform on the whole internal surface of the hole. The solution of the problem (Equation 1) and (Equation 3) is given by
(4)$$v_{z}\left(r,z\right)=-\frac{1}{j\omega \rho_{0}}\frac{\partial }{\partial z}p\left(z\right)\left[1-\frac{J_{0}\left(k_{v}r\right)}{J_{0}\left(k_{v}R_{h}\right)}\right]+j\omega \xi \left(x,y\right)\frac{J_{0}\left(k_{v}r\right)}{J_{0}\left(k_{v}R_{h}\right)},$$
where $J_{n}$ denotes the cylindrical Bessel functions of the first kind of order *n*. After relying on the approximation of the pressure derivative in a very short hole of length $h_{p}$
(5)$$\frac{\partial }{\partial z}p\left(z\right)\approx \frac{p_{inc}-p_{g}\left(x,y\right)}{h_{p}},$$
the mean value of the particle velocity over the cross-section of the hole $S_{h}=\pi R_{h}^{2}$ is
(6)$${\langle v_{z}\left(r,z\right)\rangle }_{r}=\frac{1}{S_{h}}{\;\int \int \;}_{S_{h}}v_{z}\left(r,z\right)dS_{h}\approx -\frac{1}{j\omega \rho_{0}}\frac{p_{inc}-p_{g}\left(x,y\right)}{h_{p}}F_{vh}+j\omega \xi \left(x,y\right)K_{vh},$$
with
(7)$$\begin{array}{cc} F_{vh} & =1-K_{vh},\\ K_{vh} & =\frac{2}{k_{v}R_{h}}\frac{J_{1}\left(k_{v}R_{h}\right)}{J_{0}\left(k_{v}R_{h}\right)}. \end{array}$$

The viscous force acting on the interior surface of the hole $2\pi R_{h}h_{p}$ is proportional to the normal derivative (here, ∂/∂n=−∂/∂r) of the particle velocity (Equation 4)
(8)$$F_{z}=-2\pi R_{h}h_{p}\mu {\left(\frac{\partial v_{z}\left(r,z\right)}{\partial r}\right)}_{r=R_{h}},$$
and in using (Equation 5) takes the following form
(9)$$F_{z}\left(x,y\right)\approx j\omega \xi \left(x,y\right)\Pi_{h}-\pi R_{h}^{2}K_{vh}\left[p_{inc}-p_{g}\left(x,y\right)\right],$$
with
(10)$$\Pi_{h}=2\pi R_{h}h_{p}\mu \frac{k_{v}J_{1}\left(k_{v}R_{h}\right)}{J_{0}\left(k_{v}R_{h}\right)}.$$

Dividing this force by the area associated with one hole ($4a^{2}/N$) leads to the equivalent pressure caused by the viscosity effects originating in the hole
(11)$$p_{v}\left(x,y\right)=j\omega \xi \left(x,y\right)\Pi_{h}\frac{N}{4a^{2}}-RK_{vh}\left[p_{inc}-p_{g}\left(x,y\right)\right].$$

#### 2.2.2. Wave Equation Governing the Acoustic Pressure in the Air Gap

In order to express the wave equation for the acoustic pressure $p_{g}\left(x,y\right)$ in the air gap, the following contributions of the mass per unit of time in the gap element of dimensions $dx\times dy\times h_{g}$ have to be taken into account (see velocity contributions in Figure 4):Change of the mass per unit of time in both *x* and *y* directions $-\frac{\partial }{\partial w}{\langle v_{gw}\left(w,z\right)\rangle }_{z}\;\rho_{0}\;dx\;dy\;h_{g}$, where *w* designates *x* and *y*.Contribution from the moving electrode $-j\omega \xi \left(x,y\right)\;\rho_{0}\;\left(1-R\right)\;dx\;dy$.Contribution form the holes $-{\langle v_{z}\left(r,z\right)\rangle }_{r}\;\rho_{0}\;R\;dx\;dy$, where ${\langle v_{z}\left(r,z\right)\rangle }_{r}$ is given by Equation (Equation 6).

The sum of these terms is equal to $j\omega \;{\langle \rho \rangle }_{z}\;dx\;dy\;h_{g}$ (conservation of mass) where ${\langle \rho \rangle }_{z}$ is the time-dependent acoustic density in the gap element averaged over the gap thickness. The classical solutions of linearized Navier–Stokes equation and Fourier equation for the heat conduction, under several approximations [29], give, respectively, the particle velocity and temperature variations profiles in the air gap, leading to the relations (after introducing the latter into the gas state equation) ${\langle v_{gw}\left(w,z\right)\rangle }_{z}=-\frac{1}{j\omega \rho_{0}}\frac{\partial p_{g}\left(x,y\right)}{\partial w}F_{vg}$ and ${\langle \rho \rangle }_{z}=p_{g}\left(x,y\right)\left[\gamma -\left(\gamma -1\right)F_{hg}\right]/c_{0}^{2}$ [29], with the mean values of the velocity and temperature variation profiles over the air gap thickness given by
(12)$$\begin{array}{cc} F_{vg} & =1-\frac{\tan\left(k_{v}h_{g}/2\right)}{k_{v}h_{g}/2},\\ F_{hg} & =1-\frac{\tan\left(k_{h}h_{g}/2\right)}{k_{h}h_{g}/2}, \end{array}$$
where $k_{v}$ is given by (Equation 2) and $k_{h}=\frac{1-j}{\sqrt{2}}\sqrt{\frac{\omega \rho_{0}C_{p}}{\lambda_{h}}}$. Note that these mean values are calculated for nonslip and isothermal boundary conditions at both (nonperforated) electrodes. Alternatively, the relation accounting for more realistic boundary conditions on the perforated plate [21],
(13)$$F_{(v,h)g}=1-\frac{2-R}{2}\frac{\tan\left(k_{\left(v,h\right)}h_{g}/2\right)}{k_{\left(v,h\right)}h_{g}/2},$$
can be used.

The combination of the above mentioned terms leads to the wave equation governing the acoustic pressure $p_{g}\left(x,y\right)$ in the air gap
(14)$$\left[\Delta +\chi^{2}\right]p_{g}\left(x,y\right)=-U\left(x,y\right),$$
where the source term is composed from $U\left(x,y\right)=U_{1}\xi \left(x,y\right)+U_{2}p_{inc}$ with
(15)$$\begin{array}{cc} U_{1} & =\frac{\omega^{2}\rho_{0}\left(1-RF_{vh}\right)}{F_{vg}h_{g}},\\ U_{2} & =\frac{F_{vh}R}{F_{vg}h_{g}h_{p}}, \end{array}$$
and the complex wavenumber is given by
(16)$$\chi^{2}=\frac{\omega^{2}}{c_{0}^{2}}\;\frac{\gamma -\left(\gamma -1\right)F_{hg}}{F_{vg}}-\frac{F_{vh}R}{F_{vg}h_{g}h_{p}}.$$

#### 2.2.3. Solution for the Acoustic Pressure in the Air Gap in Case of Piston-like Movement of the Moving Electrode

Since the source term *U* in (Equation 14) does not depend on the spatial coordinates x,y in this case, the solution of (Equation 14) takes the classical form
(17)$$p_{g}\left(x,y\right)=A\cos\left(\kappa_{x}x\right)\cos\left(\kappa_{y}y\right)-U/\chi^{2},$$
where $\chi^{2}=\kappa_{x}^{2}+\kappa_{y}^{2}$ (for square geometry $\kappa_{x}=\kappa_{y}=\chi /\sqrt{2}$) and *A* is an integration constant. The boundary condition is given by the acoustic pressure in the peripheral cavity (supposed to be uniform in the whole cavity volume) $p_{c}=Z_{c}w_{tot}$, where $Z_{c}$ is the acoustic impedance of the cavity and $w_{tot}$ is the total volume velocity entering to the cavity. This volume velocity is composed of the volume velocity at the output of the air gap and the volume velocity entering to the cavity through the slits
(18)$$w_{tot}=8ah_{g}{\langle v_{gw}\left(w,z\right)\rangle }_{z}-S_{s}{\bar{v}}_{s},$$
where $8ah_{g}$ is the output surface of the air gap, $S_{s}$ is the total input surface of the slits, and ${\bar{v}}_{s}≅-\frac{F_{vs}}{j\omega \rho_{0}}\frac{p_{inc}-p_{c}}{h_{p}}$ is the velocity in the slit, with $F_{vs}=1-\frac{\tan\left(k_{v}h_{s}/2\right)}{k_{v}h_{s}/2}$ being the mean value of the velocity profile through the thickness of the slit $h_{s}$ (the influence of the plate velocity on the fluid particle velocity in the slits is supposed to be negligible here). This leads directly to the boundary condition for the normal derivative of the acoustic pressure at the output of the air gap
(19)$$\partial_{n}p_{g}=-\Lambda_{c}p_{c}+\Lambda_{2}p_{inc},$$
with
$$\Lambda_{c}=\Lambda_{1}+\Lambda_{2},\;\Lambda_{1}=\frac{j\omega \rho_{0}}{8ah_{g}F_{vg}Z_{c}},\;\Lambda_{2}=\frac{S_{s}F_{vs}}{8ah_{g}h_{p}F_{vg}}.$$

The continuity of the acoustic pressure at the boundary between the air gap and the cavity can be approximately expressed using the value at the middle of the square gap side $p_{c}=p_{g}\left(a,0\right)$ (alternatively, the value at the corner $p_{g}\left(a,a\right)$ or the mean value over the side of the gap could be used). Replacing $\partial_{n}p_{g}$ and $p_{c}$ in (Equation 19) by (for example) $\partial_{x}p_{g}\left(a,0\right)$ and $p_{g}\left(a,0\right)$, respectively, and substituting from the solution (Equation 17) readily gives the integration constant
(20)$$A=\left(A_{1}\xi +A_{2}p_{inc}\right)/A_{3},$$
with
(21)$$A_{1}=\Lambda_{c}U_{1}/\chi^{2},\;A_{2}=\Lambda_{2}+\Lambda_{c}U_{2}/\chi^{2},\;A_{3}=\Lambda_{c}\cos\left(\kappa_{x}a\right)-\kappa_{x}\sin\left(\kappa_{x}a\right).$$

#### 2.2.4. Solution for the Acoustic Pressure in the Air Gap in Case of Nonuniform Movement of the Moving Electrode

Due to the symmetry of the transducer’s geometry, the solution of (Equation 14) for nonuniform U(x,y) is expressed here in the first quadrant only (namely, x,y∈(0,a)). The chosen Green’s function used in the integral formulation for the solution of (Equation 14) satisfies the same Neumann’s condition (the first derivative vanishes) at x=0,y=0 as the solution for the acoustic pressure, which can be expressed as follows [26,27,29]:(22)$$\begin{array}{cc} p_{g}\left(x,y\right)= & \int_{0}^{a}\int_{0}^{a}G\left(x,x_{0};y,y_{0}\right)U\left(x_{0},y_{0}\right)dx_{0}dy_{0}\\ & +\int_{0}^{a}\left[G\left(x,x_{0};y,a\right)\partial_{y_{0}}p_{g}\left(x_{0},a\right)-\partial_{y_{0}}G\left(x,x_{0};y,a\right)p_{g}\left(x_{0},a\right)\right]dx_{0}\\ & +\int_{0}^{a}\left[G\left(x,a;y,y_{0}\right)\partial_{x_{0}}p_{g}\left(a,y_{0}\right)-\partial_{x_{0}}G\left(x,a;y,y_{0}\right)p_{g}\left(a,y_{0}\right)\right]dy_{0}, \end{array}$$
with the Green’s function being given by
(23)$$G\left(x,x_{0};y,y_{0}\right)=g\left(x,x_{0};y,y_{0}\right)+g\left(x,-x_{0};y,y_{0}\right)+g\left(x,x_{0};y,-y_{0}\right)+g\left(x,-x_{0};y,-y_{0}\right),$$
with
(24)$$g\left(x,x_{0};y,y_{0}\right)=-\frac{j}{4}H_{0}^{-}\left(\chi \sqrt{{\left(x-x_{0}\right)}^{2}+{\left(y-y_{0}\right)}^{2}}\right),$$
where $H_{0}^{-}$ denotes the cylindrical Hankel function of the second kind of order “0”.

Taking into account the boundary condition (Equation 19), here without the slits ($\Lambda_{2}$ vanishes), solution (Equation 22) becomes
(25)$$p_{g}\left(x,y\right)=\int_{0}^{a}\int_{0}^{a}G\left(x,x_{0};y,y_{0}\right)U\left(x_{0},y_{0}\right)dx_{0}dy_{0}-p_{c}I_{g}\left(x,y\right),$$
where
(26)$$\begin{array}{cc} I_{g}\left(x,y\right)= & \Lambda_{1}\left[\int_{0}^{a}G\left(x,x_{0};y,a\right)dx_{0}+\int_{0}^{a}G\left(x,a;y,y_{0}\right)dy_{0}\right]\\ & +\left[\int_{0}^{a}\partial_{y_{0}}G\left(x,x_{0};y,a\right)dx_{0}+\int_{0}^{a}\partial_{x_{0}}G\left(x,a;y,y_{0}\right)dy_{0}\right]. \end{array}$$

The acoustic pressure in the cavity, calculated here as the mean value over the edge of the gap, $p_{c}={\langle p_{g}\left(x,a\right)\rangle }_{x}$, where ${\langle f(w)\rangle }_{w}$ denotes $\int_{0}^{a}f\left(w\right)dw/a$, can be then expressed from (Equation 25) as follows:(27)$$p_{c}=\frac{1}{1+{\langle I_{g}\left(x,a\right)\rangle }_{x}}\int_{0}^{a}\int_{0}^{a}{\langle G(x,x_{0};a,y_{0})\rangle }_{x}U\left(x_{0},y_{0}\right)dx_{0}dy_{0}.$$

### 2.3. Coupling of the Moving Electrode Displacement Field and the Acoustic Pressure Field

In this section, the strong coupling between the acoustic field inside the transducer, described in previous sections, and the displacement of the moving electrode in the form of an elastically supported rigid perforated plate and a flexible perforated plate clamped at all edges is presented.

#### 2.3.1. Elastically Supported Rigid Perforated Plate

The equation governing the displacement ξ of the elastically supported rigid plate takes the form
(28)$$\left[-M_{p}\omega^{2}+j\omega R_{p}+K_{p}\right]\xi =\int_{-a}^{a}\int_{-a}^{a}\left[p_{g}\left(x,y\right)-p_{inc}-p_{v}\left(x,y\right)\right]dxdy,$$
where $M_{p}$ is the mass of the plate, $K_{p}$ is the stiffness of the elastic support, and $R_{p}$ is the structural damping coefficient which is neglected here (all the damping in the system taken into account here originates in the acoustic fluid-filled parts of the transducer).

Reporting Equations (Equation 11) and (Equation 17) using (Equation 20) to (Equation 28) gives, after straightforward calculation, the solution for the displacement of the rigid plate:(29)$$\xi =\frac{4a^{2}\left(1+RK_{vh}\right)\left[\frac{\sin\left(\kappa_{x}a\right)\sin\left(\kappa_{y}a\right)A_{2}}{\kappa_{x}\kappa_{y}a^{2}A_{3}}-\left(1+\frac{U_{2}}{\chi^{2}}\right)\right]p_{inc}}{-M_{p}\omega^{2}+j\omega R_{p}+K_{p}+j\omega \pi N+4a^{2}\left(1+RK_{vh}\right)\left[\frac{U_{1}}{\chi^{2}}-\frac{\sin\left(\kappa_{y}a\right)\sin\left(\kappa_{y}a\right)A_{1}}{\kappa_{x}\kappa_{y}a^{2}A_{3}}\right]}.$$

#### 2.3.2. Flexible Perforated Plate Clamped at All Edges

We will depart here from the classical equation governing the displacement of the nonperforated plate [30] with the mass per unit area $M_{s}=\rho_{p}h_{p}$ ($\rho_{p}$ designates the density of the plate) and the flexural rigidity $D=\frac{Eh_{p}}{12(1-\nu^{2})}$ (*E* and ν being the Young’s modulus and Poisson’s ratio, respectively)
(30)$$\left[D\Delta \Delta -M_{s}\omega^{2}\right]\xi \left(x,y\right)=p_{g}\left(x,y\right)-p_{inc}-p_{v}\left(x,y\right),$$
clamped at all edges
(31)$$\begin{array}{cc} \xi \left(x,y\right) & =\frac{\partial }{\partial x}\xi \left(x,y\right)=0,x=\pm a,\forall y\in \left(-a,a\right),\\ \xi \left(x,y\right) & =\frac{\partial }{\partial y}\xi \left(x,y\right)=0,y=\pm a,\forall x\in \left(-a,a\right). \end{array}$$

The displacement field can be searched for in the following form of series expansion (with some truncation in practical implementation):(32)$$\xi \left(x,y\right)=\sum_{mn}\xi_{mn}\psi_{mn}\left(x,y\right),$$
where the orthonormal eigenfunctions $\psi_{mn}\left(x,y\right)$ satisfy the homogeneous equation associated with Equation (Equation 30):(33)$$\left[\Delta \Delta -k_{m,n}^{4}\right]\psi_{mn}\left(x,y\right)=0,$$
where $k_{m,n}^{4}={\left(k_{xm}^{2}+k_{yn}^{2}\right)}^{2}$. An approximate form of such eigenfunctions can be obtained as a series expansion over known functions from numerically (FEM) calculated results using the method described in [31] for nonperforated rectangular clamped plates and in [32] for perforated square clamped plates, the latter being used herein (see Appendix A).

Using the properties of the eigenfunctions [29], the modal coefficients $\xi_{mn}$ in (Equation 32) can be obtained from the relation (using Equation (Equation 11))
(34)$$\xi_{mn}\left[Dk_{m,n}^{4}-M_{s}\omega^{2}+j\omega \Pi_{h}\frac{N}{4a^{2}}\right]=\left(1-RK_{vh}\right)\int_{-a}^{a}\int_{-a}^{a}\left[p_{g}\left(x,y\right)-p_{inc}\right]\psi_{mn}\left(x,y\right)dxdy.$$

Using the relation for the acoustic pressure in the air gap $p_{g}\left(x,y\right)$ (Equation 25) along with Equations (Equation 15) and (Equation 32), Equation (Equation 34) can be expressed as follows:(35)$$\left[Dk_{m,n}^{4}-M_{s}\omega^{2}+j\omega \Pi_{h}\frac{N}{4a^{2}}\right]\xi_{mn}=c_{mn}-\sum_{qr}^{\infty }\xi_{qr}A_{\left(mn),(qr\right)},$$
or in the matrix form
(36)$$\left[B-A\right]\Xi =C,$$
where Ξ is the column vector of elements $\xi_{mn}$, B is the diagonal matrix of elements $Dk_{m,n}^{4}-M_{s}\omega^{2}+j\omega \Pi_{h}\frac{N}{4a^{2}}$, C is the column vector, and A is the matrix whose elements $c_{mn}$ and $A_{\left(mn),(qr\right)}$ are given, respectively, by
(37)$$c_{mn}=p_{inc}\int_{-a}^{a}\int_{-a}^{a}\psi_{mn}\left(x,y\right)\left\{U_{2}\left[\int_{0}^{a}\int_{0}^{a}G\left(x,x_{0};y,y_{0}\right)dx_{0}dy_{0}-MI_{g}\left(x,y\right)\right]-1\right\}dxdy,$$
and
(38)$$\begin{array}{c} A_{\left(mn),(qr\right)}=U_{1}\int_{-a}^{a}\int_{-a}^{a}\psi_{mn}\left(x,y\right)\left[\int_{0}^{a}\int_{0}^{a}G\left(x,x_{0};y,y_{0}\right)\psi_{qr}\left(x,y\right)dx_{0}dy_{0}-N_{qr}I_{g}\left(x,y\right)\right]dxdy, \end{array}$$
with
(39)$$\begin{array}{c} M=\frac{1}{1+{\langle I_{g}\left(x,a\right)\rangle }_{x}}\int_{0}^{a}\int_{0}^{a}{\langle G(x,x_{0};a,y_{0})\rangle }_{x}\;dx_{0}dy_{0},\\ N_{qr}=\frac{1}{1+{\langle I_{g}\left(x,a\right)\rangle }_{x}}\int_{0}^{a}\int_{0}^{a}{\langle G(x,x_{0};a,y_{0})\rangle }_{x}\;\psi_{qr}\left(x_{0},y_{0}\right)\;dx_{0}dy_{0}. \end{array}$$

Solving Equation (Equation 36) for Ξ gives the modal coefficients $\xi_{mn}$ and thus the displacement field of the plate ξ(x,y).

## 3. Analytical Results and Comparisons with the Numerical (FEM) Ones

In this section, the analytical results calculated using the present method are discussed and compared with the numerical (FEM) results provided by the software Comsol Multiphysics, version 6.0. The numerical formulation for the acoustic field in thermoviscous fluid inside the transducer, involving the acoustic particle velocity $\vec{v}$, acoustic temperature variation τ, and acoustic pressure *p* using the Acoustics Module [33], was coupled with the classical linear mechanical formulation for the plate provided by the Structural Mechanics Module [34]. One quarter of the transducer geometry was used for the simulation (the rest was symmetric), and the mesh consisted of tetrahedral elements combined with layered prism elements (in the boundary layers). The number of degrees of freedom varied between approximately 1 million and 3 million, depending on the dimensions of the holes in the plate (smaller holes lead to finer mesh and thus higher number of degrees of freedom). The properties of the air used in both numerical and analytical calculations are given in Table 1, and the properties of the material of the plate (silicon) are summarized in Table 2.

The displacement of the moving electrode given either by Equation (Equation 29) for the elastically supported rigid perforated plate or by Equation (Equation 32) for the flexible perforated plate clamped at all edges was used to calculate the acoustic pressure sensitivity of the electrostatic receiving transducer $\sigma =U_{0}\bar{\xi }/\left(h_{g}p_{inc}\right)$, where $\bar{\xi }=\left[{\;\int \int \;}_{S_{e}}\xi \left(x,y\right)dS_{e}\right]/S_{e}$ is the mean displacement of the plate over the surface of the backing electrode $S_{e}=4a^{2}$, and $U_{0}$ is the polarization voltage (here, $U_{0}=30$ V).

Figure 5 shows the acoustic pressure sensitivity of the receiving transducer with an elastically supported rigid perforated plate of dimensions 0.3×0.3mm (a=150μm) and thickness $h_{p}=5\;\mu m$ with N=256 square holes of side dimension $a_{h}$ varying between 0.3μm and 3μm, as per the thickness of the slits $h_{s}$. The air gap thickness is $h_{g}=4\;\mu m$, and the peripheral cavity of thickness 50μm has the volume of $V_{c}=2.72\times 10^{-12}\;m^{3}$. The mass of the plate is given by $M_{p}=\rho_{p}h_{p}\left(4a^{2}-Na_{h}^{2}\right)$, and the structural damping coefficient is supposed to be negligible $R_{p}=0\;Ns/m$. The stiffness of the elastic support $K_{p}$ was calculated from the simple numerical model of the mechanical moving part only (*in vacuo*) at very low frequencies. The dimensions of the arms of 145×30μm lead to $K_{p}=200\;N/m$. Very good agreement between the analytical results was obtained using the present method (Equation (Equation 29)), and the ones provided by the complete numerical model of the transducer can be observed, especially for small $a_{h}$ and $h_{s}$. When the values of $a_{h}$ and $h_{s}$ approach the gap thickness $h_{g}$ (Figure 5d)), the damping seems to be slightly underestimated. Generally, it seems that the “porosity” approach using the ratio R works better when the dimensions of the holes are much smaller than the gap thickness.

The acoustic pressure sensitivities of the receiving transducer with flexible perforated plate clamped at all edges of dimensions a=0.5mm and thickness $h_{p}=10\;\mu m$ with N=400 square holes of side dimension $a_{h}$ varying between 2μm and 7μm are shown in Figure 6. Here, the air gap thickness is $h_{g}=10\;\mu m$, and the volume of the cavity is $V_{c}=10^{-10}\;m^{3}$. The analytical result, calculated using the method described in Section 2.3.2, here takes into account only the first mode of the vibration of the plate $\psi_{11}$ in Equations (Equation 37) and (Equation 38), which is sufficient in the audio frequency range. Very good agreement between this analytical result and the reference numerical one can be found in the pass band of the transducer. At very high frequencies, the higher modes of the plate vibrations (not contained in the analytical results) appear in the numerical results.

At very low frequencies, the difference between the acoustic pressure in the gap $p_{g}\left(x,y\right)$ and the incident acoustic pressure $p_{inc}$ is very small due to the acoustic short circuit through the holes (see Figure 7 for $p_{inc}=1\;\mathrm{Pa}$ and hole side of 2μm (left) and 5μm (right) at f=100Hz). This leads to the numerical noise in the results of the integrals in Equations (Equation 25), (Equation 37) and (Equation 38), especially for larger holes (Figure 7b)). Since this pressure difference is the source for the plate displacement (see Equation (Equation 30)), the analytical results are perturbed by the noise in the low-frequency range (see Figure 6c,d). However, when using the transducer in the audio frequency range, the effect of the acoustic short circuit should be reduced, which leads to the use of small holes. For this case, the present analytical model gives correct results (Figure 6a)).

Using the present analytical model, the dimensions of the transducer can be further optimized, as shown in Figure 8. Smaller dimensions of the holes improve the pass band of the transducer in the lower frequency range ($a_{h}=1\;\mu m$ in Figure 8a). The dependence of the sensitivity σ on the air gap thickness $h_{g}$ in the pass band of the transducer presents the common sensitivity doubling (+6 dB) when halving the gap thickness for small holes ($a_{h}=1\;\mu m$ in Figure 8a, $a_{h}=2\;\mu m$ in Figure 8b), while in the case of larger holes, this effect almost disappears ($a_{h}=5\;\mu m$ in Figure 8c, $a_{h}=7\;\mu m$ in Figure 8d). However, the impact of decreasing $h_{g}$ on increasing damping of the resonance, which is usual in condenser microphones, is preserved in the case of a perforated moving electrode. Note that the thickness of the plate $h_{p}$ influences the mass and stiffness of the plate, hence the resonance frequency and amplitude of the plate displacement. Higher thickness $h_{p}$ leads to higher resonance frequency (thus, higher pass band of the transducer) and lower sensitivity.

## 4. Conclusions

The analytical model of an electroacoustic transducer with the moving electrode in the form of a perforated plate (rigid elastically supported or flexible clamped at all boundaries) was developed. The formulation for the acoustic pressure field in the air gap between the moving and the fixed electrode was derived, taking into account the acoustic short circuit through the holes, the thermal and viscous boundary layers effects in the thin fluid film, and the coupling with the displacement of the plate. The displacement of the plate and the acoustic pressure sensitivity of the transducer used as a microphone was calculated, and the latter was compared to the reference numerical (FEM) results. Very good agreement between these models was found in the transducer pass band, and some discrepancies, appearing generally out of the frequency range of interest, were discussed and explained. The influence of some geometrical parameters of the transducer, such as dimensions of the holes in the plate or air gap thickness, was investigated.

Note that only the first mode of the flexible plate vibration was taken into account here using the analytically expressed approximation of its first eigenfunction calculated numerically. This is sufficient in the audio frequency range; however, further research should focus on improved expression of the eigenfunctions, providing better results at the frequencies above the first resonance, where the higher modes of the perforated plate vibration occur.

## Figures and Tables

**Figure 1 micromachines-14-00921-f001:**
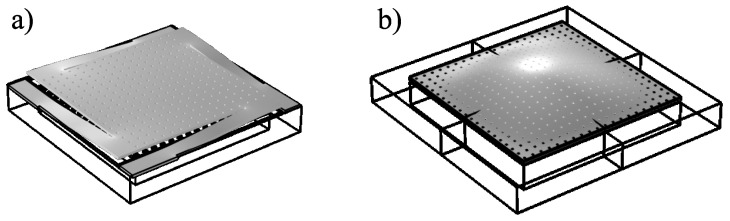
MEMS transducers with perforated moving electrodes in the form of (**a**) the rigid elastically supported square plate and (**b**) the flexible square plate clamped at all boundaries.

**Figure 2 micromachines-14-00921-f002:**
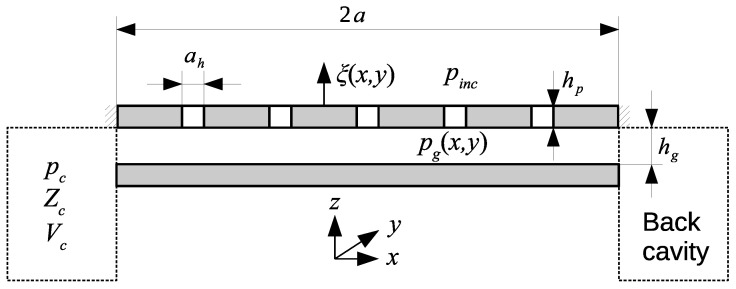
Sketch of the whole system.

**Figure 3 micromachines-14-00921-f003:**
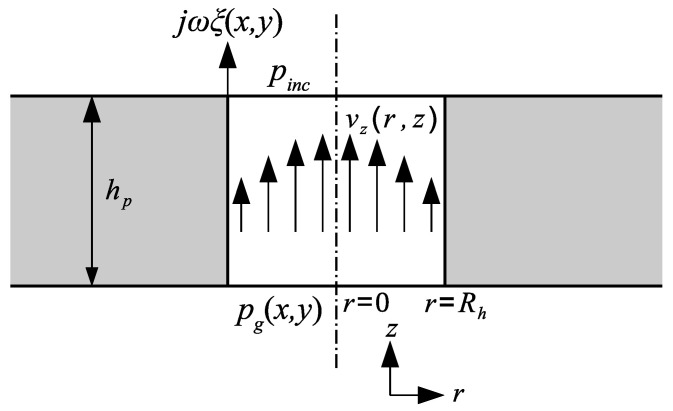
Sketch of the hole in the moving electrode.

**Figure 4 micromachines-14-00921-f004:**
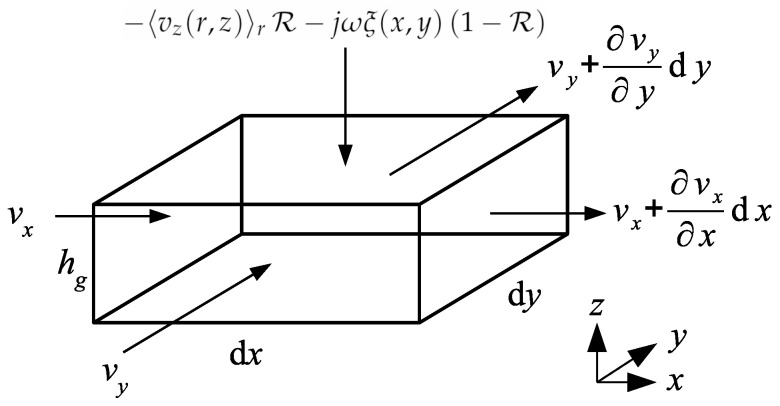
Element of the air gap.

**Figure 5 micromachines-14-00921-f005:**
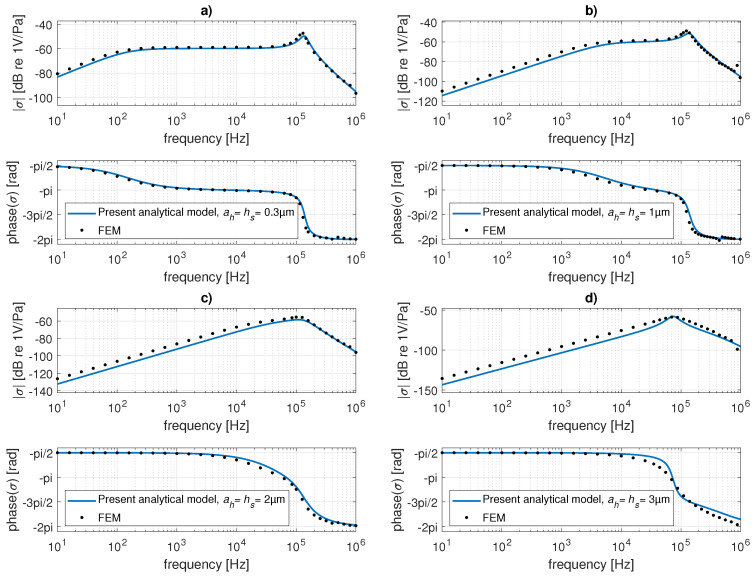
Magnitude (upper curves) and phase (lower curves) of pressure sensitivity of the transducer with elastically supported perforated plate: comparison of the present analytical results (continuous lines) with the numerical (FEM) result (black points) for the side of the holes and the thickness of the slits being equal to (**a**) 0.3 μm, (**b**) 1 μm, (**c**) 2 μm, and (**d**) 3 μm.

**Figure 6 micromachines-14-00921-f006:**
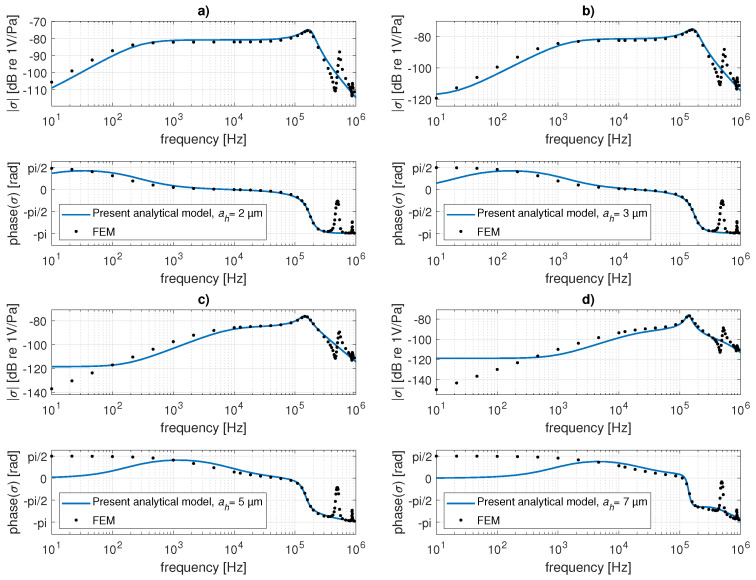
Magnitude (upper curves) and phase (lower curves) of pressure sensitivity of the transducer with flexible perforated clamped plate: comparison of the present analytical results (continuous lines) with the numerical (FEM) result (black points) for the side of the holes being equal to (**a**) 2 μm, (**b**) 3 μm, (**c**) 5 μm, and (**d**) 7 μm.

**Figure 7 micromachines-14-00921-f007:**
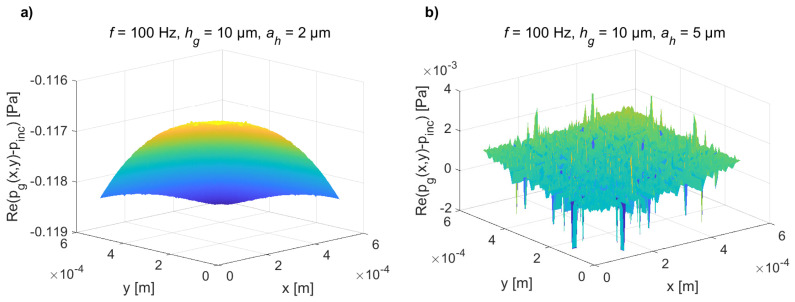
Real part of the difference of the acoustic pressure between both sides of the plate $ℜ\left[p_{g}\left(x,y\right)-p_{inc}\right]$ in the first quadrant (x,y∈(0,a)) calculated using the present method at f=100Hz for $p_{inc}=1\mathrm{Pa}$ and for the side of the holes being equal to (**a**) $a_{h}$ = 2 μm (left) and (**b**) $a_{h}$ = 5 μm (right).

**Figure 8 micromachines-14-00921-f008:**
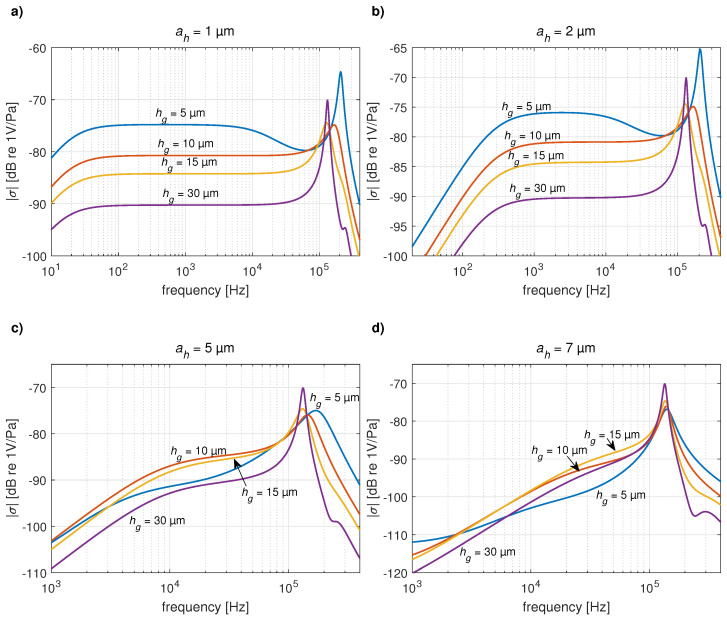
Magnitude of pressure sensitivity of the transducer calculated using the present method: the effect of varying air gap thickness $h_{g}$ for the side of the holes being equal to (**a**) 1 μm, (**b**) 2 μm, (**c**) 5 μm, and (**d**) 7 μm.

**Table 1 micromachines-14-00921-t001:** Properties of the air.

Parameter	Value	Unit
Adiabatic sound speed $c_{0}$	343.2	m s${}^{-1}$
Air density $\rho_{0}$	1.2	kg m${}^{-3}$
Shear dynamic viscosity μ	$1.814\times 10^{-5}$	Pa s
Thermal conductivity $\lambda_{h}$	$25.77\times 10^{-3}$	W m${}^{-1}$ K${}^{-1}$
Specific heat coefficient at constant pressure per unit of mass $C_{p}$	1005	J kg${}^{-1}$ K${}^{-1}$
Ratio of specific heats γ	1.4	-

**Table 2 micromachines-14-00921-t002:** Material properties of the plate (silicon).

Parameter	Value	Unit
Plate density $\rho_{p}$	2330	kg m${}^{-3}$
Young’s modulus *E*	160	G Pa
Poisson’s ratio ν	0.27	-

## Data Availability

Not applicable.

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
