# Peer review of "Modeling of MEMS Transducers with Perforated Moving Electrodes"

_micromachines, 2023, doi:10.3390/mi14050921_

Round 1

Reviewer 1 Report

This paper discusses modeling of electroacoustic transducer with perforated moving plates. The acoustic pressure field has been modeled in the air gap between fixed and moving electrodes while considering acoustic short circuit through the holes The analytical model's results are comparable to the FEM data obtained through COMSOL. The influence of geometrical parameters has also been studied. The model works well for frequencies of interest. The paper is comprehensive and has merit to be accepted for publication.

Reviewer 2 Report

1. Authors need to highlight the manuscript's novelty using a table and compare it with state-of-the-art literature. 

2) In the analytical calculations it is suitable to include the effect of the diameter of the hole, which can be attractive to understanding the microfabrication limitations. 

3) In the analytical calculations it is suitable to include the effect of the hp, which can be attractive from the microfabrication side, regarding choosing the suitable silicon wafer.

4) Authors should include relevant literature such as

i) https://www.nature.com/articles/s41598-022-19693-5  ii)https://ieeexplore.ieee.org/abstract/document/9774849 iii)https://onlinelibrary.wiley.com/doi/full/10.1002/admi.202202446 
